# Socioeconomic Inequities in Youth Participation in Physical Activity and Sports

**DOI:** 10.3390/ijerph18136946

**Published:** 2021-06-29

**Authors:** Pooja S. Tandon, Emily Kroshus, Katharine Olsen, Kimberly Garrett, Pingping Qu, Julie McCleery

**Affiliations:** 1Center for Child Health, Behavior and Development, Seattle Children’s Research Institute, Seattle, WA 98105, USA; ekroshus@uw.edu (E.K.); kim.garrett@seattlechildrens.org (K.G.); Pingping.Qu@seattlechildrens.org (P.Q.); 2Department of Pediatrics, University of Washington, Seattle, WA 98195, USA; 3Center for Leadership in Athletics, University of Washington, Seattle, WA 98195, USA; kmolsen@uw.edu (K.O.); juliem4@uw.edu (J.M.)

**Keywords:** physical activity, sports, children, inequity, affluence

## Abstract

Access to opportunities for physical activity and sports, and therefore potential benefits of participation, are distributed inequitably. The aims of this study were to describe and compare youth experiences related to sport and physical activity by socioeconomic factors. A cross-sectional survey was conducted of students in 5–12th grades in King County, Washington, USA. Students were asked about physical activity and sports experiences and about demographic factors including family affluence, which was categorized as low, medium, and high. Participants were 1038 youth (50% girls, 58% non-White, and 32% from homes where languages other than English are spoken). Children from low-affluence families reported fewer days/week of physical activity, fewer sports sampled, and lower rates of ever playing sports. Greater proportions of children from low-affluence families reported these barriers to sports: (1) don’t want to get hurt, (2) don’t feel welcome on teams, (3) too expensive, and (4) transportation. Middle school children from high-affluence families had three times higher odds of meeting physical activity recommendations, and high-affluence high schoolers had three times higher odds of ever participating in sports compared to peers from low-affluence families. Socioeconomic status was inversely associated with outcomes related to youth physical activity and sports participation. The disproportionately reported barriers to sports participation are modifiable, and cross-sector solutions can help promote play equity.

## 1. Introduction

Physical activity in youth contributes to a range of positive outcomes including reduced risk of obesity and improved cardiometabolic health, mental health, and academic performance [1,2,3]. However, only 24% of U.S. children meet physical activity guidelines of at least 60 min/day of moderate-to-vigorous physical activity [4]. School-based physical education, organized sport and physical activity programs, unstructured free play, and general outdoor recreation provide a variety of ways for youth to be physically active [5]. However, access to these opportunities, and therefore potential benefits of participation, are distributed inequitably. There are documented inequities in youth physical activity, with youth from lower income families and those living in lower resource communities engaging in less physical activity [6]. Organized sport participation and sport specialization [7] tend to be lower in children from families of lower socioeconomic status [8]. There are fewer commercial facilities for physical activity and fewer options for organized sport in communities characterized by a high proportion of low-income residents [9]. Affluent neighborhoods often have more pedestrian- and bike-friendly facilities, are safer from crime and traffic, have a more favorable appearance, and host better access to recreation facilities than low-income neighborhoods [10]. Black, Indigenous, and people of color (BIPOC), as well as lower income families, are less likely to have access to a nearby park, and the parks in their neighborhoods tend to be smaller and of lower quality [11,12].

Simultaneously, an industry has sprung up around physical activity provision. A highly structured youth sports model has commodified organized physical activity: instead of free play and outdoor unstructured activity, youth are funneled into paid programming. This model is inaccessible to many youth for reasons including high cost, transportation, and location of programming [13,14,15]. This trend likely makes it more challenging for some youth to engage in sport sampling—the process of children trying out various sports and physical activities—which is considered a developmentally appropriate means for building physical literacy and limiting burnout and injuries [16]. There is limited data currently on how sport sampling and sports attrition are patterned by socioeconomic factors [17].

Although research suggests that a combination of neighborhood, household, and individual factors can explain socioeconomic inequalities in sports participation and physical activity [18], previous studies tend to focus on the relationship between physical and social environmental factors with physical activity [19,20]. Household-level factors such as family affluence and English language proficiency of parents could be critical determinants of children’s access to opportunities to try various sports and sustain participation. Furthermore, understanding individual-level factors from youth perspectives, such as their own perceived barriers to participation, would provide insight into modifiable solutions to address them [21]. The objectives of this study are to describe and compare youth experiences and attitudes related to sport and physical activity by socioeconomic status.

## 2. Methods

### 2.1. Sample and Procedure

Survey data were collected as part of the State of Play: Seattle-King County study from students in 5th through 12th grade from 14 Schools in King County, Washington (USA). They were stratified into twelve groups by grades served (elementary, middle, and high schools) location (urban/rural), and income (low-income schools were defined as schools where more than 40% of the students received free- or reduced-price lunch). Schools were selected from within each strata, oversampling for low-income schools, resulting in twenty-four randomly selected schools. Ultimately, only three school districts agreed to the research, and only five schools from the originally-selected sample agreed to participate. A Community Advisory Board engaged three additional schools for a total of eight schools and distributed the youth survey to participants in local after-school programs in alignment with the strata outlined above. Parents provided passive permission for their child to participate in the survey. This study was approved by the Institutional Review Board at the University of Washington.

### 2.2. Measures

#### 2.2.1. Demographic and Socioeconomic Factors

Questions related to Race/ethnicity and language(s) spoken at home were taken directly from the 2018 Washington State Healthy Youth Survey (HYS) [22]. Affluence was determined using the Family Affluence Scale, an age-appropriate scale for assessing the socioeconomic status of children and adolescents [23]. Children and adolescents typically do not have accurate information on their family’s finances, so a less intrusive, more comprehensible approach has to be utilized to identify their socioeconomic status. These questions were asked: (1) Does your family own a car, van, or truck? (2) Do you have your own bedroom for yourself? (3) During the past 12 months, how many times did you travel away with your family? Responses were divided into Low (3–5), Medium (6–7), and High (8–9) affluence for analyses.

#### 2.2.2. Physical Activity and Sports Participation

Participants reported physical activity by recalling the number of days they were physically active for at least 60 min per day in past seven days, which is the same question used in the HYS [22]. Park use was assessed with one survey item: “Think about the outdoor parks and green spaces that are closest to your home. In the past year, during the warmer months, how often did you play at these outdoor parks and green spaces?” Number and type of sports sampled were assessed with a survey item sourced from the Aspen Institute’s Sport and Society Program’s State of Play landscape analyses nationwide and edited by the research team to enhance clarity and inclusivity based on stakeholder feedback. Participants were asked to select the activities they had tried in the last year from a list of 47 sports and add others not listed. Respondents self-reported sport participation on an organized sports team or organized athletic activity, which was defined as a group that meets on a regular basis and is led by a coach or instructor.

#### 2.2.3. Barriers

Question about perceived barriers for different types of physical activity were developed using a stakeholder-engaged process in which questions created by the research team were reviewed by diverse stakeholders from the State of Play: Seattle-King County Advisory Board for content and clarity. Barriers to participation in organized sports were assessed with two survey items:(1)For those respondents who have never played organized sports: If you don’t play organized sports, what are the reasons why? Select up to three.(2)For those respondents who played organized sports before but stopped: What are the reasons why you stopped playing or participating? Select up to three.

### 2.3. Analysis

All data were analyzed using R version 4.0.3. Continuous measures were summarized using means and standard deviations, and categorical variables using frequencies and proportions. Logistic regression models were used to determine the odds ratios of meeting physical activity standards and participating in organized sports based on various respondent characteristics.

## 3. Results

Participants were 1038 youth in grades 5–12 from King County, WA. Half of the respondents were girls, 58% were non-White, and 32% were from homes where languages other than English were spoken. Using the family affluence scale, 13% of respondents were categorized as coming from families with “low affluence”, 31% as “medium”, and 56% as “high” affluence. Table 1 has more details about participant characteristics.

Physical activity and sports participation by demographic characteristics and family affluence are shown in Table 2. On average, boys reported more days per week of physical activity (4.41 vs. 3.97) and more of them met the recommendation of 60 min of daily physical activity (23% vs. 16%). Rates of sports participation ever and in the past year were similar between boys and girls in this sample. Compared to youth from homes where only English is spoken, youth coming from homes where no English is spoken reported lower rates of ever playing organized sports (57% vs. 90%) and had sampled fewer sports (11.1 vs. 13.0). Children from low-affluence families reported fewer days per week of participation in physical activity (3.5 vs. 3.76 medium vs. 4.46 high), fewer sports sampled (8.45 vs. 10.20 vs. 14.77), lower rates of ever having played organized sports (63% vs. 79% medium vs. 80% high), or playing in sports in the past year (69% vs. 73% medium vs. 84% high).

For children that reported never having played organized sports or having quit playing, barriers to participation by affluence level are reported in Table 3. The top three most commonly reported barriers were (1) lack of interest in sports, (2) not having time due to schoolwork, and (3) not feeling good enough to play, and there were no difference by affluence level for these responses. There were statistically significant differences by affluence level for the next several barriers with greater proportions of children from low-affluence families reporting that they (1) don’t want to get hurt (35%), (2) don’t feel welcome on teams (27%), (3) find sports too expensive (27%), and (4) don’t have a way to get to practices/games (18%). Twenty-three percent of children from low-affluence families also reported that they don’t have time to play sports due to family responsibilities. No participants reported that safety concerns at fields/gyms/courts were a barrier to sports participation for them.

For the middle school students (grades 5–8), children in higher grades had lower odds (OR 0.67; CI 0.52, 0.86) of meeting the physical activity recommendation of 60 min per day. Children from more affluent families had higher odds of meeting recommendations (OR 3.08, CI 1.07, 8.87) compared to the low-affluence families. For high school students (grades 9–12), boys had higher odds (OR 1.81, CI 1.02, 3.22) than girls of meeting physical activity recommendations. Other demographic characteristics and family affluence did not result in statistically significant differences. The results for children who reported participating in organized sports in the past year trended towards significance for meeting recommendations (OR 2.76; CI 0.93, 8.2; *p* = 0.067). The full results from the regression model are presented in Table 4.

Table 5 displays the results from the regression model regarding the odds of children ever having participated in organized sports. For middle school children, there were no statistically significant results although the odds ratios were in the hypothesized directions. For high school students, children from medium-affluence families had twice the odds (OR 2.08; CI 1.06, 4.11) and those from high-affluence families had more than thrice the odds (OR 3.33; CI 1.62, 6.85) of ever having participated in sports compared to children from low-affluence families. Lower odds ratios for sports participation were seen for youth who identified as Hispanic, Black, and Asian, with the results for the last two groups reaching statistical significance. Children from families where no English is spoken at home had lower odds of ever having participated in organized sports with statistically significant results for the middle school age group (OR 0.27; CI 0.14, 0.53).

## 4. Conclusions

Using data reported by youth, this study found that socioeconomic status is positively associated with meeting physical activity recommendations and sports participation among children in grades 5 through 12. During the middle school years, children from medium and high-affluence families were about three times as likely to meet physical activity recommendations compared to children from low-affluence families, even when controlling for sports participation and time spent in parks. For high school students, although family affluence did not emerge as an independent predictor of meeting physical activity recommendations, sports participation was significantly lower for children from lower affluence families. Previous evidence on the relationship between SES and physical activity has been mixed, with lower SES frequently associated with children’s sedentary time and obesity, but not consistently with lower physical activity [24,25,26,27]. While we also did not find consistent results across the age groups in our sample, our findings suggest areas of concern related to equitable access and opportunities for children to engage in physical activity. Poverty has been shown to have a detrimental effect on child health with lifelong consequences [28], and these results extend the evidence to show that lower affluence is also associated with children’s participation in optimal amounts of health promoting physical activity. Consistent with considerable literature that finds boys are more likely to meet physical activity recommendations than girls, sex was also a statistically significant predictor in our analyses.

For high school students, family affluence was positively associated with the likelihood of ever trying organized sports. In the entire sample, only two-thirds of children from low-affluence families had ever tried an organized sport, compared to 89% from high-affluence families. In terms of sports participation, family affluence was a significant factor for the high school students but not middle school age children. This could reflect differences in participation rates in general between younger and older children or represent differences in access and costs related to extracurricular sports (school and community based) for children at different ages.

For high school students, students from families with medium and high affluence were twice and thrice as likely, respectively, to have ever participated in sports compared to children from low-affluence families. Of note, children from families where no English is spoken at home had significantly lower odds of ever having participated in organized sports across all grades and even when family affluence was accounted for in the models. For immigrant youth, participation in extracurricular activities may be particularly beneficial in terms of strengthening connections with their school and community [29], and the reasons for non-participation may be complex [30]. These findings point to notable inequities in access to opportunities for children who may already be marginalized in our current system due to coming from families with fewer resources and/or those that face language barriers.

Youth-reported barriers to physical activity can help us understand these inequities [14], While concerns about time, skills, and interest in sports were common across youth from all affluence levels, certain additional barriers were more likely to be reported by youth from less affluent families. Approximately one third of children from low-affluence families endorsed worries about getting hurt, not feeling welcome on teams, and/or that sports are too expensive. Fewer than 17% of children from medium- and high-affluence families reported that any of these were concerns for them. Sports organizations at recreational and competitive levels, including those that are school-based, need to be aware of these reported barriers and consider strategies to help mitigate them. This includes ensuring that all youth are not only able to access opportunities for sport and physical activity to allow for developmentally appropriate sport sampling, but that they feel safe and welcome in these spaces. Feeling welcome can mean many things; some youth may feel less-qualified for school sports teams. No-cut school sports teams can continue to be vehicles for all youth to participate in a team experience, engage in physical activity, and strive for personal achievement. Others may not feel welcome due to their individual differences from the majority group. Coach training focused on how to lead equitably while recognizing and mitigating unconscious bias is imperative. Further, supporting people of color, women, LGBTQ, and people of all abilities to be coaches can diversify youth physical activity leadership and in turn can help youth feel more welcome on teams. Eighteen percent of children from low-affluence families also reported transportation barriers, which were rare for the other groups of children. This is another modifiable barrier with potential solutions including free access to public transportation for youth, considering access to parks and community centers in long-term transportation planning, increasing safety measures on public transit, and creating more safe walking and biking routes to parks and playfields.

This study has several limitations that should be considered. Although we had a large sample with a high proportion of youth from non-white and non-English speaking families, it was from one county in Washington State, which limits its generalizability. Self-report data are also subject to recall and social desirability biases. In addition, there may be some misclassification of family affluence given that it was collected by report from children. We did use validated and reliable scales, when possible, to try to mitigate some of these concerns.

In conclusion, socioeconomic status, operationalized as child-reported family affluence in our study, was inversely associated with outcomes related to physical activity and sports participation. In addition, children from families where English is not spoken at home were also less likely to have ever participated in sports. The barriers to sports participation reported by children, particularly those disproportionately endorsed by children from lower affluence families, are modifiable. Targeted solutions can support access for these marginalized groups, including collaborations between community-based organizations that serve immigrant youth, public agencies, educational institutions, and private-sector organizations. Collaborations of this sort require public and private organizations to prioritize increased youth physical activity as an important individual and community health outcome. Funding, policy support, and multisectoral collaboration are necessary to develop an intentional set of community-based strategies that eliminate barriers related to cost, transportation, and language, which could go a long way to increasing equitable access to sports for more children.

## Figures and Tables

**Table 1 ijerph-18-06946-t001:** Participant characteristics.

Characteristic (*n* = 1038)	*n* (%)
Gender	
Female	502 (50%)
Grade	
5	223 (22%)
6	110 (11%)
7	100 (10%)
8	47 (5%)
9	287 (28%)
10	65 (6%)
11	146 (14%)
12	59 (6%)
Race/Ethnicity	
Asian or Asian-American	265 (26%)
Black or African–American	71 (7%)
Hispanic or Latino/Latina	80 (8%)
White or Caucasian	437 (42%)
Two or more races	145 (14%)
Other races	35 (3%)
Language(s) spoken at home	
English only spoken at home	702 (68%)
English and other languages spoken at home	192 (18%)
No English spoken at home	144 (14%)
Family Affluence	
Low (score of 3–5)	125 (13%)
Medium (score of 6–7)	291 (31%)
High (score of 8–9)	523 (56%)

**Table 2 ijerph-18-06946-t002:** Physical activity and sports participation by demographic and socioeconomic factors.

CHARACTERISTIC	Physical Activity Days per Week of 60 min/day Mean (SD)	Meeting PA Recommendation (60 min/day × 7 days per week) *n* (%)	Number of Sports ever Sampled Mean (SD)	Ever Played Organized Sports *n* (%)	Played Organized Sport in Last Year *n* (%)
Gender					
Female	3.97 (2.14)	78 (16%)	13.19 (7.96)	390 (83%)	314 (81%)
Male	4.41 (2.14)	111 (23%)	11.99 (8.38)	371 (82%)	288 (78%)
Grade					
5–8	4.58 (2.05)	115 (25%)	15.54 (8.08)	380 (83%)	321 (85%)
9–12	3.84 (2.17)	79 (15%)	10.11 (7.37)	406 (82%)	302 (75%)
Language(s) at home					
English only spoken at home	4.35 (2.11)	147 (21%)	13.00 (8.14)	585 (90%)	460 (79%)
English and other languages spoken at home	3.94 (2.11)	31 (17%)	12.33 (7.69)	128 (74%)	106 (83%)
No English spoken at home	3.65 (2.26)	16 (12%)	11.10 (8.7)	74 (57%)	58 (79%)
Family Affluence					
Low affluence	3.5 (2.33)	15 (12%)	8.45 (6.74)	72 (63%)	49 (69%)
Medium affluence	3.76 (2.19)	43 (15%)	10.20 (7.12)	214 (79%)	156 (73%)
High affluence	4.46 (2.02)	110 (22%)	14.77 (7.94)	455 (89%)	380 (84%)

**Table 3 ijerph-18-06946-t003:** Barriers to sports participation (among those who do not currently participate in organized sport) by family affluence (*n* = 316).

Barriers	Total	Affluence (Low, Score 3–5)	Affluence (Medium, Score 6–7)	Affluence (High, Score 8–9)	*p*-Value
I am not interested in sports	140 (44%)	30 (45%)	50 (43%)	60 (45%)	0.911
I don’t have time to play sports due to schoolwork	129 (41%)	22 (33%)	44 (38%)	63 (47%)	0.111
I’m not good enough to play	113 (36%)	27 (41%)	46 (39%)	40 (30%)	0.194
I don’t want to get hurt	61 (19%)	23 (35%)	18 (15%)	20 (15%)	**0.002**
I don’t feel welcome on sports teams	55 (17%)	18 (27%)	15 (13%)	22 (17%)	**0.044**
Sports are too expensive	49 (16%)	18 (27%)	16 (14%)	15 (11%)	**0.011**
I don’t have time to play sports due to family responsibilities	45 (14%)	15 (23%)	13 (11%)	17 (13%)	0.08
I don’t have a way to get to practices/games	31 (10%)	12 (18%)	11 (9%)	8 (6%)	**0.025**
I don’t know what sports are available in my community	32 (10%)	9 (14%)	11 (9%)	12 (9%)	0.566
I don’t feel safe at the fields, gyms, or courts	0 (0%)	0 (0%)	0 (0%)	0 (0%)	1

Bold *p*-Values signify statistical significance at *p* < 0.05.

**Table 4 ijerph-18-06946-t004:** Characteristics Associated with Meeting the Physical Activity Standard.

Characteristic		Grade 5 to 8 OR 95 CI	Grade 5 to 8 *p* Value	Grade 9 to 12 OR 95 CI	Grade 9 to 12 *p* Value
Grade		0.67 (0.52, 0.86)	0.002	1.07 (0.82, 1.38)	0.622
Gender	Female	-	-	-	-
	Male	1.44 (0.89, 2.35)	0.139	1.81 (1.02, 3.22)	0.044
Race	White	-	-	-	-
	Black	2.09 (0.84, 5.17)	0.111	0.49 (0.06, 4.06)	0.506
	Hispanic or Latino/Latina	1.58 (0.6, 4.15)	0.356	1.36 (0.42, 4.43)	0.61
	Asian or Asian-American	0.69 (0.35, 1.35)	0.283	0.52 (0.19, 1.44)	0.208
	Other races	1.05 (0.33, 3.3)	0.934	1.38 (0.14, 13.49)	0.781
	Two or more races	1.36 (0.68, 2.75)	0.387	2.42 (1.17, 5.01)	0.018
Affluence	Low (score of 3–5)	-	-	-	-
	Medium (score of 6–7)	2.74 (0.91, 8.2)	0.072	0.65 (0.23, 1.85)	0.423
	High (score of 8–9)	3.08 (1.07, 8.87)	0.037	0.96 (0.36, 2.56)	0.936
Language spoken at home	English spoken at home	-	-	-	-
	No English spoken at home	0.85 (0.36, 1.99)	0.704	0.26 (0.03, 2.07)	0.202
Park time	Never	-	-	-	-
	Ever	1.77 (0.48, 6.46)	0.389	0.92 (0.34, 2.49)	0.873
Organized Sport Participation	No	-	-	-	-
	Yes	1.9 (0.87, 4.19)	0.109	2.76 (0.93, 8.2)	0.067

**Table 5 ijerph-18-06946-t005:** Characteristics Associated with ever having participated in organized sports.

Variable		Grade 5 to 8 OR 95 CI	Grade 5 to 8 *p* Value	Grade 9 to 12 OR 95 CI	Grade 9 to 12 *p* Value
Grade		1.09 (0.82, 1.45)	0.539	1.06 (0.83, 1.34)	0.656
Gender	Female	-	-	-	-
	Male	1 (0.57, 1.77)	0.987	1.23 (0.74, 2.05)	0.432
Race	White	-	-	-	-
	Black	0.62 (0.18, 2.14)	0.452	0.28 (0.11, 0.75)	0.011
	Hispanic or Latino/Latina	0.56 (0.17, 1.85)	0.346	0.55 (0.21, 1.42)	0.218
	Asian or Asian-American	0.43 (0.18, 1)	0.05	0.5 (0.26, 0.99)	0.046
	Other races	0.39 (0.11, 1.41)	0.151	1.16 (0.12, 10.94)	0.895
	Two or more races	0.62 (0.22, 1.72)	0.355	0.92 (0.38, 2.21)	0.854
Affluence	Low (score of 3–5)	-	-	-	-
	Medium (score of 6–7)	1.02 (0.43, 2.42)	0.969	2.08 (1.06, 4.11)	0.034
	High (score of 8–9)	1.64 (0.71, 3.8)	0.249	3.33 (1.62, 6.85)	0.001
Park time	Never	-	-	-	-
	Ever	1.85 (0.7, 4.87)	0.211	1.3 (0.64, 2.66)	0.469
Language spoken at home	English spoken at home	-	-	-	-
	No English spoken at home	0.27 (0.14, 0.53)	0	0.51 (0.24, 1.09)	0.083

## Data Availability

The data presented in this study are available on request from the corresponding author.

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
