# Peer review of "Socioeconomic Inequities in Youth Participation in Physical Activity and Sports"

_ijerph, 2021, doi:10.3390/ijerph18136946_

Round 1
Reviewer 1 Report
This study aiming to describe the economic factors related to physical activity and sports participation in the young population from Washington, US.
This paper explains in a positive way when looking at the topic of adolescents health and strategies to improve their daily physical activity. I think some things need clarifying for the publication that will help the futures results.
Comment 1: Line 34 – Please add a reference.
Comment 2: Line 64 – Please add a reference.
Comment 3: I suggest the authors use the Numbered style for the references in the text. Sometimes is confusing to read.
General Comment: Congratulations to the authors for this work. This is manuscript is clear and well written.
Author Response
Comment 1: Line 34 – Please add a reference.
Response 1: reference added
Comment 2: Line 64 – Please add a reference.
Response 2: reference added
Comment 3: I suggest the authors use the Numbered style for the references in the text. Sometimes is confusing to read.
Response 3: We have changed the reference style and hopefully this is clearer now.
Reviewer 2 Report
Thank you for making the time to conduct this meaningful and valuable study. This is the kind of study that many people have a "general sense" but with no data about. The data really encourage related parties to do something about it.
While the paper at this stage has shown the conditions of school children’s involvement or not in sports and structured activities, would it be possible to include a few suggestions as to how the current situation can be addressed? A few recommendations specifically referring to the inadequacies of current planning and provisions will be a good idea to conclude the paper.
Author Response
Comment 1: While the paper at this stage has shown the conditions of school children’s involvement or not in sports and structured activities, would it be possible to include a few suggestions as to how the current situation can be addressed? A few recommendations specifically referring to the inadequacies of current planning and provisions will be a good idea to conclude the paper.
Response 1: We offer some recommendations in lines 265-283 and 296-303. We have also added some additional content in lines 320-328.